# Treatment of Wastewater from a Grass Carp Pond with Multiple-Batch Forward Osmosis by Using Sucrose as a Draw Solution

Yuliang Xu †, Xia Cheng † and Jianghui Du *

School of Biosystems Engineering and Food Science, Zhejiang University, Hangzhou 310058, China
* Correspondence: jianghui.du@foxmail.com
† These authors contributed equally to this work.

**Abstract:** Forward osmosis (FO), a green and economical membrane technology driven by a natural concentration gradient, has attracted increasing attention for wastewater treatment because it consumes less energy and removes large amounts of pollutants. In this research, an approach based on an FO mechanism that could improve the concentration of organic pollutants in wastewater collected from grass carp ponds was proposed. The wastewater serving as a feed solution (FS) was concentrated by FO, and sucrose was used as a draw solution (DS). The multiple chemical oxygen demand (COD) variation, the water flux, and the reverse solute flux during the FO process were investigated. The results indicated that the water flux and the reverse solute flux had similar trends in the processing of batch experiments 1–8, and the concentrating multiple of organic contaminants reached a maximum of 3.5 in the whole study. In addition, membrane fouling was studied via a scanning electron microscope (SEM), and a loose cake layer was deposited on the membrane surface. Moreover, findings from energy dispersive scanning (EDS) analysis showed that the fouling substances in the support layer of the membrane were mainly organic compounds and silica. In contrast, the dominant contaminants of the active layer contained several microelements (such as K and Ca) in addition to organic compounds. Three-dimensional (3D) fluorescence analysis confirmed that the FS components could enter the DS and the chemical components of the sucrose solution could also enter the FS. The findings of this study provide a new view on selecting a DS and protecting the aquaculture environment.

**Keywords:** forward osmosis; aquaculture wastewater; draw solution; sucrose solution; membrane fouling; multiple batches

## 1. Introduction

The grass carp breeding industry is an indispensable component for developing the freshwater fishery economy [1], playing an important role in providing essential animal protein for human beings. Pond cultivation is a significant part of the recirculating pond system for breeding grass carp fish. In recent years, the effect of aquaculture wastewater treatment has become an issue of concern [2]. Pollutants contained in aquaculture water mainly include fish excreta, residual food, and the remains of aquatic animals and plants [3–5]. Residual food not entirely consumed by fish and dissolved in water could become an especially major source of aquaculture water pollution [6,7]. These residual pollutants in the aquaculture water cause water eutrophication and present a challenge to the survival of aquaculture animals. Moreover, when microorganisms metabolize and decompose these residual foods, the rapid consumption of oxygen in the water leads to the anoxic death of aquatic organisms. Traditional aquaculture wastewater treatment technologies include sedimentation [8,9], constructed wetland [10,11], ecological ditch [12,13], and bio-filtration [14,15]. However, these methods usually require a large area, leading to low treatment efficiency.

Membrane filtration is a physical filtration process and is rarely involved in complex chemical reactions, so it performs better in operational applications. Meanwhile, a membrane filtration system with small pore size functions to concentrate liquids because it allows selective permeation of liquid components [16–19]. Traditionally, membrane-based separation processes for treating aquaculture wastewater are either thermal or pressure driven. Membrane distillation (MD) is able to remove organic matter, inorganic matter, and microalgae from aquaculture wastewater. However, it relies on heat generated by input energy, which may prevent its large-scale application [20]. Nanofiltration (NF) and reverse osmosis (RO) are pressure-driven processes, and these technologies may be subject to a relatively high capital cost and operational energy cost [21]. In summary, an aquaculture wastewater treating technology needs to be developed that is low cost, is convenient to operate, and relies less on external conditions.

Because of the low pollutant concentration and large output, grass carp wastewater should be concentrated by a membrane system to improve treatment efficiency. As a newly developed membrane filtration method in recent years, FO consumes less energy and has lower operational cost, and there is lighter membrane fouling compared with the traditional pressure-driven membrane filtration system [22,23]. During FO operation, one side of the FO membrane is the feed solution, containing pollutants of low concentration, while the other side is a DS, with a high-concentration solution. In an FO system, driven by the difference in osmotic pressure between FS and DS, pure water is able to transport through a semipermeable membrane from the FS into the DS [24,25]. Since the membrane rejects solute molecules and ions, the reduction in the concentration of the DS and the increase in contaminant concentration in the FS with the absorption of pure water from the FS to the DS achieves the dual purposes of recovering water resources from the FS and concentrating them. Recent advances in FO treatment have been used in desalination, wastewater treatment, food processing, regional cooling, and power generation [26,27]. Farid et al. combined nanobubbles and FO systems to treat aquaculture wastewater, whereas nanobubbles served as a physical membrane-cleaning agent that enhanced the performance of the FO membrane. This hybrid system shows great potential in treating small volumes of wastewater; however, it would require a nanobubble generating infrastructure and electric energy to ensure operation [28]. Until now, the application of FO in aquaculture wastewater using sucrose as a DS has been rarely reported. Generally, the wastewater from freshwater ponds has a low concentration, demanding a large treatment capacity for wastewater treatment.

The motivation for the FO system mainly derives from the fact that it relies on natural osmotic pressure rather than external energy input. Thus, applying the FO system could reduce cost. Furthermore, the fouling layer on the FO membrane surface could be easily removed by some physical methods (such as hydraulic flushing) because this fouling layer has no compacted structure in comparison with that produced under the conditions of pressure-driven membrane technologies [29–31]. For instance, over 90% of the water flux fouling the FO membrane could be recovered with the application of hydraulic cleaning [30]. Furthermore, hydraulic flushing could also clean the fouled FO membrane, polluted during municipal sewage treatment, with flux recovery reaching 49.37% [32]. Grass carp wastewater treatment via the FO system could, therefore, achieve two goals: concentrating aquaculture wastewater and recovering water resources. Concentrated aquaculture wastewater could help by both improving the efficiency of the subsequent treatment unit and reducing the treatment capacity.

During the membrane treatment process, it is crucial to control membrane fouling since it is directly related to FO performance and membrane lifetime when applying membrane systems in wastewater treatment [33]. Compared with pressure-driven membrane technology, the driving force of the FO process is generated by the difference in osmotic pressure between the two sides of the permeability membrane, which should reduce cost and lower membrane pollution [34]. However, the wastewater from real applications typically contains many organic pollutants, inorganic pollutants, and microorganisms, which

will attach to the surface of the FO membrane under the effect of physical, chemical, and biological reactions. The accumulation of fouled materials could cause a decrease in water flux and affect the service life of the membrane. Furthermore, membrane fouling is highly determined by the composition of wastewater and the membrane orientation, significantly impacting water flux and the recovery of water resources [35,36]. Therefore, to mitigate membrane fouling, it is essential to find out the characteristics of FO surface adhesion.

To enhance the working efficiency of the FO system, selecting an appropriate DS is critical. Sucrose is soluble in water and is an organic substance with a large molecule size. Thus, a DS formulated by sucrose could effectively reduce the negative influence from reverse solute osmosis in the FO process. In addition, the diluted sucrose solution could be applied as a carbon source to be put into the pond and enhance the heterotrophic bacterial activity [37]. Meanwhile, adding various carbohydrate sources, especially simple carbohydrates (such as sucrose), to the aquaculture water could reduce the concentration of nitrogen compounds in the pond by increasing the C/N ratio and thereby increasing the activity of microorganisms and reducing the pH of the water. Therefore, adding a sucrose DS would improve the survival rate of aquatic products and growth performance [38].

Grass carp ponds produce a large amount of wastewater with a low concentration of pollutants. An appropriate method is necessary to increase the concentration of pollutants in the wastewater and enhance the efficiency of treating this kind of wastewater. The idea in this work comes from a common fishery application where a sucrose solution serves as a carbon source. In this study, real wastewater from a grass carp pond was treated by an FO system, in which sucrose solution was employed as a DS, to recover pure water and increase the concentration of the organic pollutants in grass carp wastewater. The diluting sucrose solution could simultaneously be employed as a carbon source to be added to the fish pond. The performance of FO in treating aquaculture wastewater was investigated in terms of COD concentration, water flux, and reverse solute flux. The influence of physical cleaning on the recovery of membrane flux was also studied. Moreover, the characteristics of membrane fouling and the main composition of the fouling layer were investigated after treatment. To the best of our knowledge, this is the first study that has researched the possibility and potential challenges of pond aquaculture wastewater treated by FO with the sucrose solution as the DS. This work may provide a potential method for improving the treating efficiency of real aquaculture wastewater.

## 2. Materials and Methods

### 2.1. FO Membrane and Experiment Device

Figure 1 displays the schematic diagram of the FO system used in this work. The FO membrane (8040 CTA 85 SDS, FTS, Hydration Technology Inc., Albany, NY, USA) includes an active layer and a supportive layer. The supportive layer was embedded into the active layer to enhance the mechanical strength of the membrane. The main composition of the membrane is cellulose triacetate, the pH tolerance of this membrane is 4–9, the water contact angle is 65°, and the thickness is about 50 μm. Furthermore, the FO membrane is preserved in DI water at 0–4 °C and the FO device should be cleaned before being employed in experiments.

The key part of the FO device is the membrane filtration unit, which consists of two membrane blocks, and the FO membrane is installed in the middle of the two blocks. The active layer of the FO membrane faces the DS and the supportive layer faces the FS. The effective area of the FO membrane is 25 cm$^2$, and each membrane block is 15 cm in length, 15 cm in width, and 5.3 cm in depth. Each membrane block has 10 precision channels (0.42 cm in width, 5 cm in length, and 0.5 cm in depth). Two peristaltic pumps (BT600-2J, Baoding Longer Precision Pump Inc., Baoding, China) are used to drive independent circulation of the FS and the DS on both sides of the membrane, where the FS and the DS circulate in a cross-flow direction in each membrane room. Note that the stream directions of the FS and the DS in the membrane filtration unit are opposite each other, and the membrane fouling and the reverse solute flux could be reduced [39]. Furthermore, to calculate water

flux, the weight change of the FS was monitored by an electronic balance (ACS-W-3kg, Shanghai Yingzhan Electromechanical Inc., Shanghai, China). The temperature of the DS was stabilized using a thermostat water bath cauldron (HH-1, Changzhou Tianrui Instrument Co., Ltd., Changzhou, China).

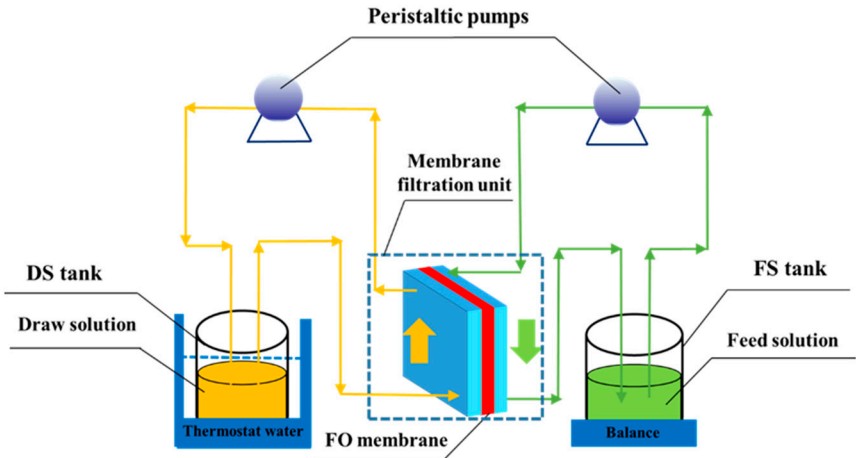

**Figure 1.** Schematic diagram of the FO experimental setup.

### 2.2. Feed Solution and Draw Solution

As shown in Figure 2, the real aquaculture wastewater in this study was sampled from a large-scale grass carp pond belonging to a commercial aquaculture company located in Hangzhou, China. About 2400 grass carps were cultured in this pond. The mass percentages of the main components of the fish feed are as follows: rice bran 40%, wheat bran 38%, soya bean cake 10%, fish meal 10%, and yeast 2%. This fish feed applied in the grass carp production was obtained from Tongwei Fishing Feed Inc. (1038 jetsam, Chengdu, China), and the feeding time for these fishes was 9:00–10:00 a.m. and 3:00–4:00 p.m. To mitigate the negative influence of suspended solid pollutants and other large-size insoluble compounds, it is necessary to conduct pre-treatment before formal FO experiments. In this research, the aquaculture wastewater had to be filtered using filter paper before FO treatment. Moreover, due to the unstable concentration of organic contaminants caused by adding fish feeds, the sampling time of the real wastewater was about 8:00, so the sampling time was not within the range of the feeding time.

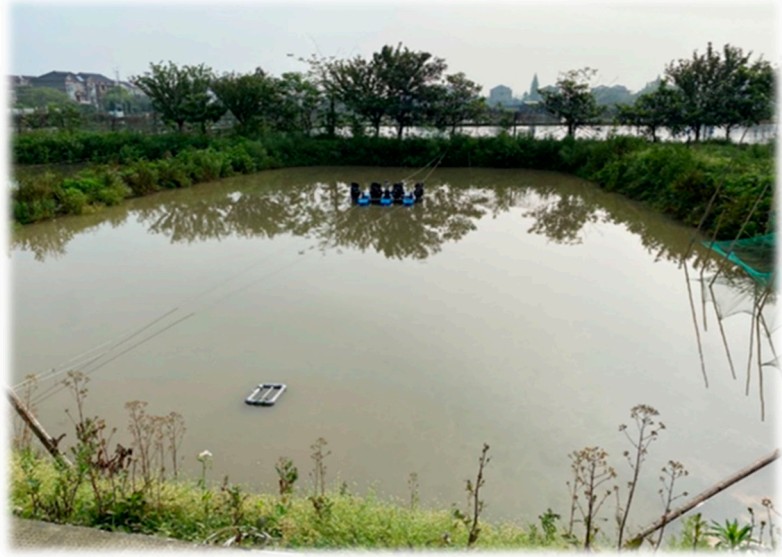

**Figure 2.** Picture of real grass carp pond.

The DS was prepared by dissolving sucrose (AR, Macklin Inc., Shanghai, China) in DI water at room temperature. The initial volume of the FS and the DS was adjusted to 1 L before carrying out the experiment.

After filtration, the pH of the real pond wastewater is in the range of 7.4–7.9, electrical conductivity is 1150–1300 μS/cm, dissolved oxygen is ~7.6 mg/L, and the temperature is nearly 15 °C. The concentration of COD and $NH_4^+$-N in pond wastewater is in the range of 40–70 mg/L and <1 mg/L, respectively. In this regard, the organic pollutants dissolved in water are the most significant contributors to the formation of the cake layer of the membrane because most insoluble suspended matter could be removed by a pre-treatment process, such as filtration via filter paper. For the $NH_4^+$-N, an extremely low concentration could be detected in water samples. From this perspective, the negative effect of $NH_4^+$-N in the FO process could not severely interfere with the final results of this work.

### 2.3. Experimental Procedures

In our previous works, the DS was formulated by sucrose and the dynamic variation in the COD concentration in the FS and the average water flux of the FO system were studied with synthetic aquaculture wastewater made by grass carp feed as the FS. The performance of FO was also researched on the basis of selected operational factors, such as the DS concentration, the cross-flow velocity, and the temperature of the DS. The response surface methodology (RSM) model related to COD concentration and average water flux was also established based on single-factor experiments and CCD response surface methodology. According to the findings from the RSM experiments, the optimized operation parameters of the FO process were determined as follows: 2 mol/L of sucrose solution, 18.7 cm/s of cross-flow velocity, and 33.6 °C of DS. The above operational conditions were applied to carry out the treatment of this real wastewater by FO.

The whole FO experiment in this work was divided into 20 dependent running cycles, in which a running cycle was 180 min. According to our previous membrane orientation testing, the installation model of FO in membrane blocks was that the DS faced the active layer. In this way, a larger water flux could be obtained. Distilled water was used to physically clean the polluted FO membrane after the FO system had worked for 24 h (8 running cycles). Afterward, the cleaned FO membrane was continually used to conduct subsequent experiments. For physical cleaning, both DS and FS were replaced with distilled water (1 L for volume) and the membrane was cleaned at a cross-flow velocity of 60 cm/s for 30 min. In the process of an experimental cycle, the changes in the weight of the FS were recorded every 10 min to calculate the average water flux. After the 20 working cycles were completed, the membrane morphology of the FO active layer and the support layer was determined by scanning electron microscope and energy dispersive scanning (EDS–SEM) analysis.

### 2.4. Analytic Methods

2.4.1. Water Flux Calculation

The water flux could be applied to evaluate water recovery efficiency during the wastewater process by FO technology. In this study, the water flux was calculated by the weight variation in FS and the weight data of the FS was recorded every 10 min, leading to varied water flux values. Therefore, the average value of water fluxes recorded in one running cycle was selected to assess the water permeability of the FO membrane. Water flux is calculated using the following equation [40]:

$$J_w = \frac{(m_2 - m_1)}{\rho(t_2 - t_1)A_m} \tag{1}$$

where $J_w$ is water flux (L·m$^{-2}$·h$^{-1}$), $m_1$ is the weight of the FS at time $t_1$ (kg), $m_2$ is the weight of the FS at time $t_2$ (kg), $A_m$ is the effective membrane area (m$^2$), and $\rho$ is pure water density (kg·m$^{-3}$).

### 2.4.2. Chemical Oxygen Demand and Change Multiples

The concentration of organic contaminants in the FS can be assessed by measuring the chemical oxygen demand (COD). A significant change in the COD concentration of the FS is observed after it goes through the FO system's concentrating process because the membrane intercepts macromolecular pollutants. Therefore, in this work, the change multiples of COD in the FS were employed to represent the efficiency of organic pollution treatment during the FO process. The procedures of the COD testing method were as follows: (1) The wastewater samples from the FS were placed in a COD digestion instrument (DRB200, HACH Co., Ltd., Loveland, CO, USA) and digested at a high temperature (150 °C) for 120 min. (2) The digested samples were cooled to room temperature. (3) The COD concentration was detected by a spectrophotometer (DR2800, HACH Co., Ltd., Loveland, CO, USA). Initial and final COD concentrations were measured in a single running cycle to calculate change multiples. The COD change multiples were used to determine the treatment efficiency of organic contamination, according to Formula (2).

$$N = \frac{C_{t,Fs}}{C_{0,Fs}} \qquad (2)$$

where $N$ represents the change multiples, $C_{0,Fs}$ is the COD concentration in the FS at the beginning of one running cycle (mg·L$^{-1}$), and $C_{t,Fs}$ is the COD concentration in the FS at the end of the running cycle (mg·L$^{-1}$).

### 2.4.3. Reverse Solute Flux

During the real application of the FO mechanism, water molecules could cross the semipermeable membrane and various pollutants were intercepted, achieving the recovery of pure water from the FS. However, the solute in the DS could also transfer into the FS due to the concentration differences on both sides of the membrane, which could lead to a negative impact on the enhancement of FO efficiency. Formula (3) was used to calculate the reverse solute flux [41].

$$J_s = \frac{C_t V_t - C_0 V_0}{\Delta t A_m} \qquad (3)$$

where $J_s$ is the reverse solute flux (g·m$^{-2}$·h$^{-1}$); $C_0$ and $C_t$ are the initial and final COD concentrations of the FS, respectively (mg·L$^{-1}$); $V_0$ and $V_t$ are the initial and final volumes of the FS, respectively (L); $\Delta t$ is the working time of FO treatment (h); and $A_m$ is the effective membrane area (m$^2$).

### 2.4.4. Analysis of Membrane Morphology and Pollutants

To study the formation process of the cake layer, the characteristics of the fouled membrane were analyzed. The membrane fouling sample was prepared with the experimental membrane taken from the FO module, and it was naturally dried in glassware (12310, Labselect, Hefei, China) for 24 h. Generally, membrane characterization includes different dimensions, such as membrane thickness, membrane morphology, and the composition of pollutant elements. These characteristics were investigated via scanning electron microscope and energy dispersive scanning (EDS–SEM) (SU-8000, Hitachi, Tokyo, Japan) [42,43]. In real aquaculture wastewater, the composition of organic pollutants is complex and three-dimensional fluorescence spectroscopy (F4500, HITACHI, Japan) can be used to characterize the variations in dissolved organic compounds in both DS and FS. For the three-dimensional fluorescence spectroscopy test, the DS and FS were sampled at the initial and ending stage of all batch experiments. The data from three-dimensional fluorescence spectroscopy were analyzed through the Origin Pro 2017 according to the method of fluorescence region integration.

## 3. Results and Discussion

### 3.1. Detection for Water Index during Experiments

Unlike the aquaculture facilities established in the room, the water quality and operational factors of the natural fish pond are highly influenced by weather conditions and environmental variation. It is necessary to detect the water parameters related to the aquaculture products' health to assess the effect of the water quality of the pond during FO treatment and reduce the uncertainty of pre-treatment before formal FO treatment.

In this research, the quality of the fish pond water was observed for about 10 days. As shown in Figure 3a, the pH of the wastewater was in the range of 7.5–7.9, indicating that the water was neutral. Under this pH, the water was suitable for the growth and reproduction of living fish and other microorganisms. Moreover, low concentrations of $NO_2^-$-N (0.01–0.04 mg/L) and $NH_4^+$-N (0.1–0.6 mg/L) could be obtained, as shown in Figure 3b,c, and ammonia nitrogen and nitrite nitrogen at these levels do not influence the normal physiological metabolic activity of aquaculture animals [44]. Ammonia is a critical water index and its amount in aquaculture water must be controlled since, at high concentrations, it adversely influences aquaculture survival [45–47]. Another crucial environmental factor that cannot be ignored for the production of outside aquaculture breeding is the weather, which is highly dependent on the different seasons, that is, rainfall, temperature, and sunshine. In this work, the temperature of the water body was unstable, fluctuating in the range of 12–19 °C (Figure 3d) because the experiment was performed in spring (April 2022), when the weather changes are dramatic. In fact, the temperature was of significance for the metabolic reaction of microorganisms and other aquaculture, leading to a change in the living environment of aquaculture organisms and animal health [48]. Temperature can also influence the growth of aquatic plants, influencing the ecosystem of the pond. Hence, the variation in the composition of various pollutants in wastewater plays an important role in the effect of pre-treatment of the FO process, and monitoring the quality indexes of water is essential in real production.

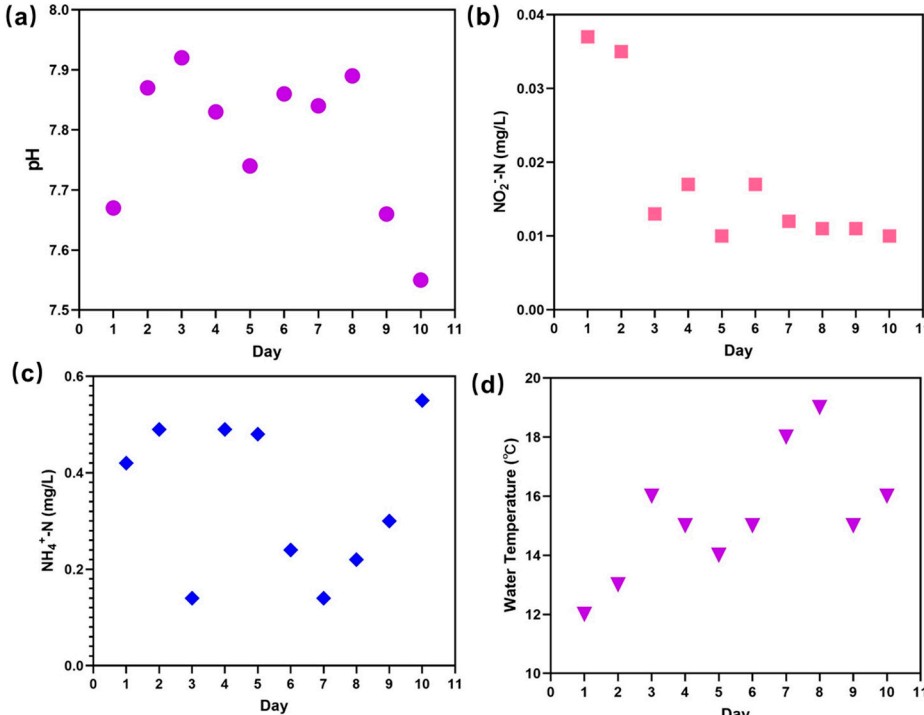

**Figure 3.** Monitoring the water quality of a grass carp pond: the pH of the wastewater (**a**); the concentration of $NO_2^-$-N and $NH_4^+$-N (**b**,**c**); and the temperature of the water body (**d**).

*3.2. FO Performance*

3.2.1. Concentration and Multiples of COD

Original and finishing COD concentrations in the FS tank were measured in every batch experiment during the FO treatment of the grass carp aquaculture wastewater. The change multiples, which were calculated using the COD concentration in 20 batch experiments, are shown in Figure 4. For batch experiments 1–8, the COD change multiples decreased from the highest, 3.5 times, to 1.9 times, indicating that membrane fouling had increased with the extension of working time during the FO process. Moreover, the pollutants attached to the membrane surface led to a decrease in membrane flux and COD change multiples. When batch experiments 1–8 were completed (the red dotted line in Figures 4 and 5), the FO membrane was physically cleaned by replacing the FS and the DS with distilled water. The 9th to 16th batch experiments were carried out after cleaning the membrane under a cross-flow velocity at 60 cm/s for 30 min. The COD change multiples reduced from 2.73 times to 1.87 times after the 9th to 16th batch experiments. Furthermore, compared with the COD change multiples of the 8th and 9th batches of the FS, it was discovered that the change multiples increased from 1.9 times to 2.73 times after physical cleaning, indicating that physical cleaning could recover the water flux of the FO membrane system and improve the concentrating efficiency of the FO membrane. However, the COD multiples could not recover to the initial level, at which the concentrating multiple was 3.5 times. In addition, at the end of the 16th batch, the device was also physically cleaned and the COD change multiples were recovered to a certain degree.

3.2.2. Water Flux and Reverse Solute Flux

Figure 5 displays the results of water flux and reverse solute flux in 20 batch experiments. During batch experiments 1–8, the water flux decreased from 25.01 $L \cdot m^{-2} \cdot h^{-1}$ to 16.85 $L \cdot m^{-2} \cdot h^{-1}$, showing a trend similar to that of COD concentration in the same running cycle. One reason for the decline in the water flux was the accumulation of pollutants on the membrane surface during the operation of the FO system, and the presence of a fouled layer led to increased mass transfer resistance and decreased water molecular permeability. Concentration polarization generated from the structural factors of the membrane also caused additional resistance during membrane separation, reducing the average water flux during the FO process [42]. The polluted membrane was cleaned after the eighth batch testing, when the water flux increased to 20.64 $L \cdot m^{-2} \cdot h^{-1}$, but the overall water flux trend gradually declined. During experimental cycles 1–8, the trend of reverse solute flux and water flux was basically the same: the reverse solute flux increased with the increase in water flux, and the reverse solute flux decreased with the decrease in water flux. With increased membrane fouling, the level of interception of organic matter by the FO membrane gradually increased [43], which might have caused the gradual thickening of the fouling layer on the membrane surface at the later stage of system operation, resulting in the reduced permeability of organic macromolecules and reduced water flux.

In addition, as demonstrated in Table 1, the water flux and the reverse solute flux achieved in this study were compared with those in other FO systems for treating wastewater from various sources. It is found that the filtration performance of the membrane and the leakage solute of the DS are highly dependent on the type of target wastewater and operational conditions, such as the material used to fabricate the membrane and its structural characteristics, classifications and concentration of the DS, and the dominant pollutant of real wastewater. Moreover, the tendency of reverse solute flux can also increase with the increase in the DS concentration, although the water flux can be enhanced simultaneously. This phenomenon might be related to the total amount of solute dissolved in the DS because the high concentration of the DS could raise the concentration polarization of the FO process to a certain degree, posing a challenge to the normal mass transfer process. Thus, in the future, research on controlling the dynamic balance of enhancing DS concentration and decreasing the reverse solute flux is necessary.

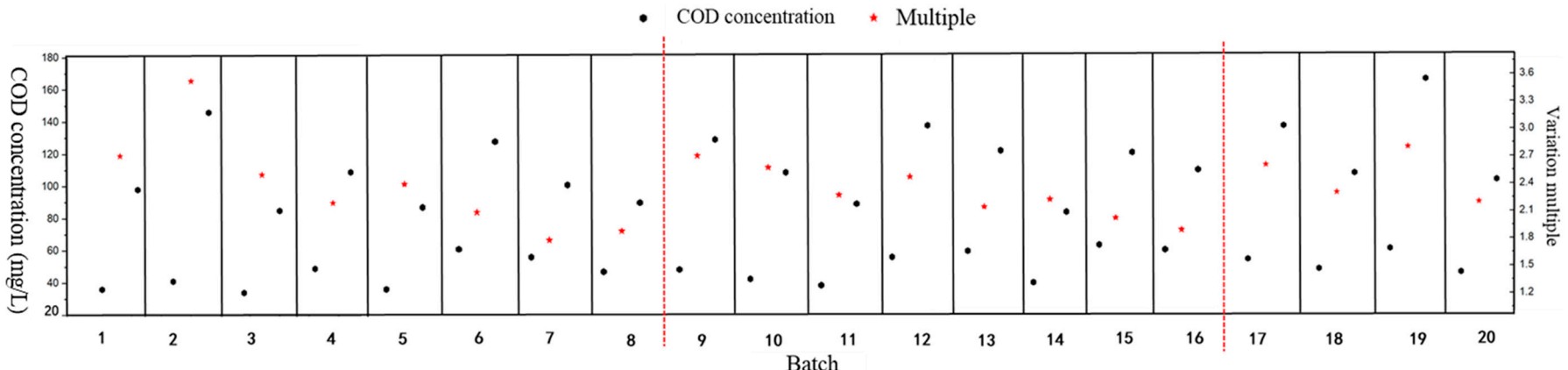

**Figure 4.** Concentration and change multiples of COD: In a single running batch, the black dot above is for effluent COD and the black dot below is for influent COD.

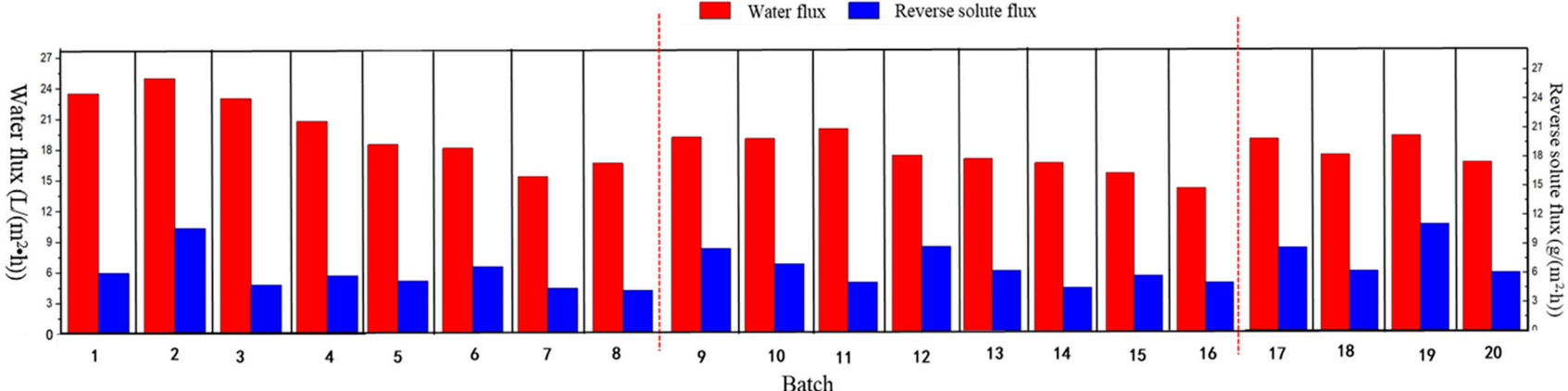

**Figure 5.** Water flux and reverse solute flux in 20 working batches.

**Table 1.** Water flux and reverse solute flux in an FO system for wastewater treatment.

| Feed Solution | Draw Solution | Water Flux (L/(m$^2$·h)) | Reverse Solute Flux (g/(m$^2$·h)) | Reference |
|---|---|---|---|---|
| Aquaculture wastewater | 2 mol/L sucrose | 15.4–25.3 | 4.3–11.6 | This work |
| Textile wastewater | 1.5 mol/L Na$_2$SO$_4$ | 18.6 ± 0.4 | 5.1–8 | [49] |
| Dairy wastewater | 1 mol/L NaCl | 10–18 | 2–5 | [50] |
| Tetracycline wastewater | 2 mol/L NaCl | 12–18 | 4–12 | [51] |
| Landfill leachate | 0.5 mol/L NaCl | 3.11–5.84 | 4.62–5.33 | [52] |

### 3.3. Analysis of Membrane Pollutants

3.3.1. Microscopic Morphology of Pollutants on the Membrane Surface

SEM was used to observe the morphology of the cake layer on the membrane and analyze the characteristics of the fouling membrane during wastewater treatment via the FO process [53]. The surface of the support layer of the polluted membrane was covered with pollutants, making the surface texture fuzzy. The polluted layer presents a lamellar structure with an irregular shape, an uneven distribution, and large surface roughness; many irregular bulges and granular objects can be clearly seen (Figure 6a,b). It has been reported that the initial stage of membrane fouling formation is mainly the adsorption between membrane and pollutants, ensuring that the pollutants adhere to the membrane surface. After the formation of the fouling layer, the interaction between pollutants became the main driving factor for the thickening of the fouling layer [54].

In this work, the active layer of the membrane faced the DS and the texture structure could be clearly observed due to fewer pollutants on the membrane active layer (Figure 6c,d). Cake layer formation was affirmed via SEM photos of the cross section (Figure 6e,f). The cake layer was formed because there were a large number of suspended solids and other pollutants in the aquaculture wastewater of grass carp. Some cracks could also be seen on the membrane surface, formed possibly due to the drying process of the membrane. Moreover, a small amount of particulate matter was present on the surface of the active layer, which might have been formed by a part of the solute in the DS remaining on the membrane surface. Previous studies have found that humic acid, protein, and polysaccharides are the main components of the pollution layer during FO filtration of wastewater [55]. Membrane fouling can reduce the effective area of the membrane and decrease the effective osmotic pressure of the FO system, and the organic pollutants can also increase the membrane resistance to a certain extent [56]. Therefore, to reduce membrane fouling in FO treatment, efficient methods for pre-treatment should be explored to purify the FS.

3.3.2. EDS Analysis of the Membrane Active Layer

Figure 7 displays the EDS analysis of the fouled active layer of the FO membrane after the 20 batch experiments are completed. In this study, the active layer of the FO membrane faces the DS. The EDS analysis demonstrated that, in the new membrane, there were only two elements, C and O, predominantly related to the manufacturing materials of the membrane (Figure 7a,b). However, when it comes to membrane fouling, the percentage of different elements was remarkably different. The main elements of the active layer contaminants were C and O, indicating that the main pollutions were organic substances. Moreover, the contents of C and O in the membrane were the highest and their total mass percentage was more than 90%, which was decided by the membrane material. A small number of inorganic elements, such as Ca and K, were also detected on the surface of the membrane active layer (Figure 7c,d), which might have been caused by some substances in the FS penetrating the membrane to cross over to the DS side.

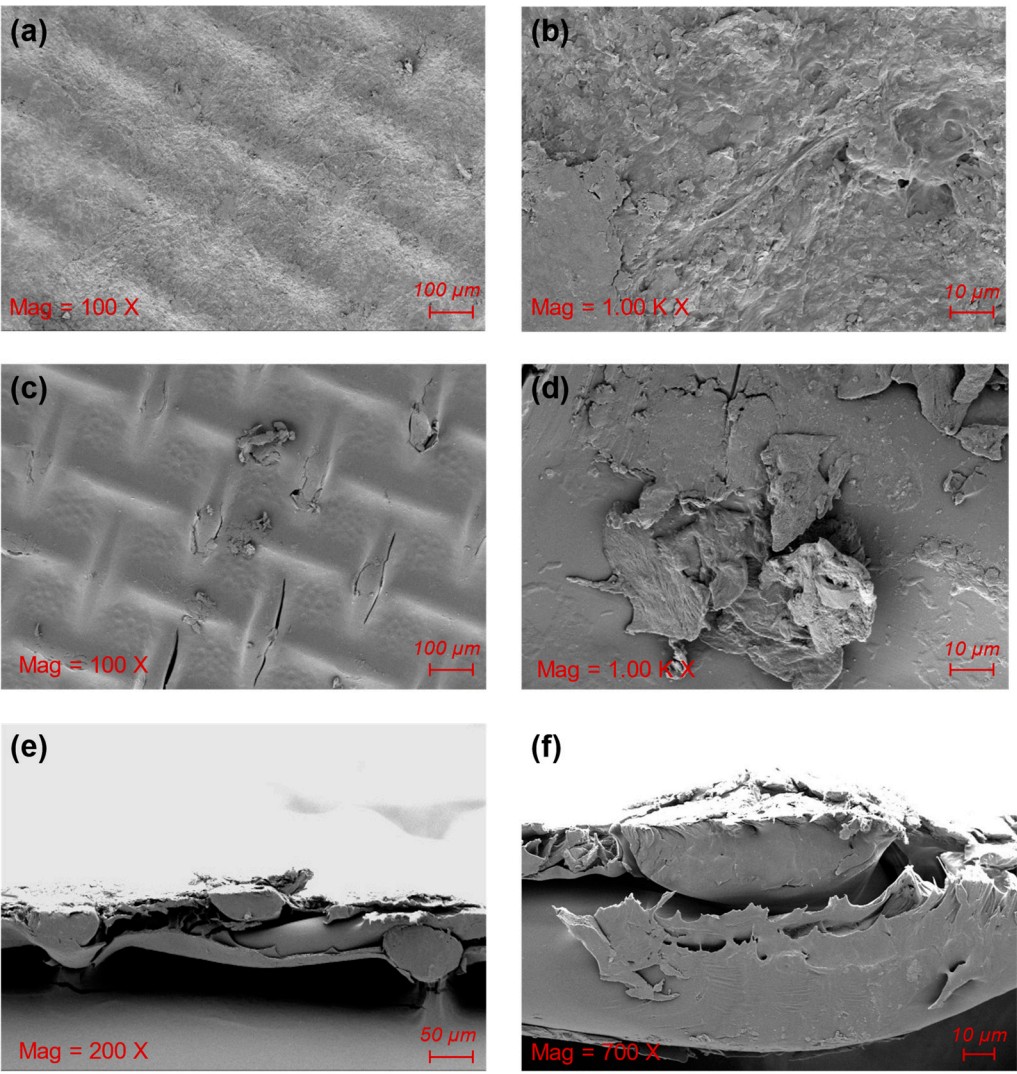

**Figure 6.** SEM photos of the polluted CTA membrane: the polluted support layer (**a**,**b**); the polluted active layer (**c**,**d**); and the cross section of the polluted membrane (**e**,**f**).

Although the number of inorganic elements in the FS was low, a water-insoluble inorganic salt formed easily [57] and further accumulated on the surface of the forward osmosis membrane to reduce the filtration performance of the membrane. The SEM scanning analysis and the EDS elements analysis of the membrane support layer are shown in Figure 7e,f, respectively. The membrane support layer faced the FS. The main pollutants of the support layer were C, O, and Si, confirming that the fouling components were mainly organic compounds and silicon dioxide. Small amounts of Al, Mg, Ca, K, and other trace inorganic elements were detected in addition to the elements C and O. The accumulation of pollutants on the membrane surface could reduce the membrane pores, resulting in decreased membrane permeability. The inorganic salt ions in the solution also increased the fouling tendency of the membrane since both anions and cations could easily combine to form water-insoluble crystalline substances [58].

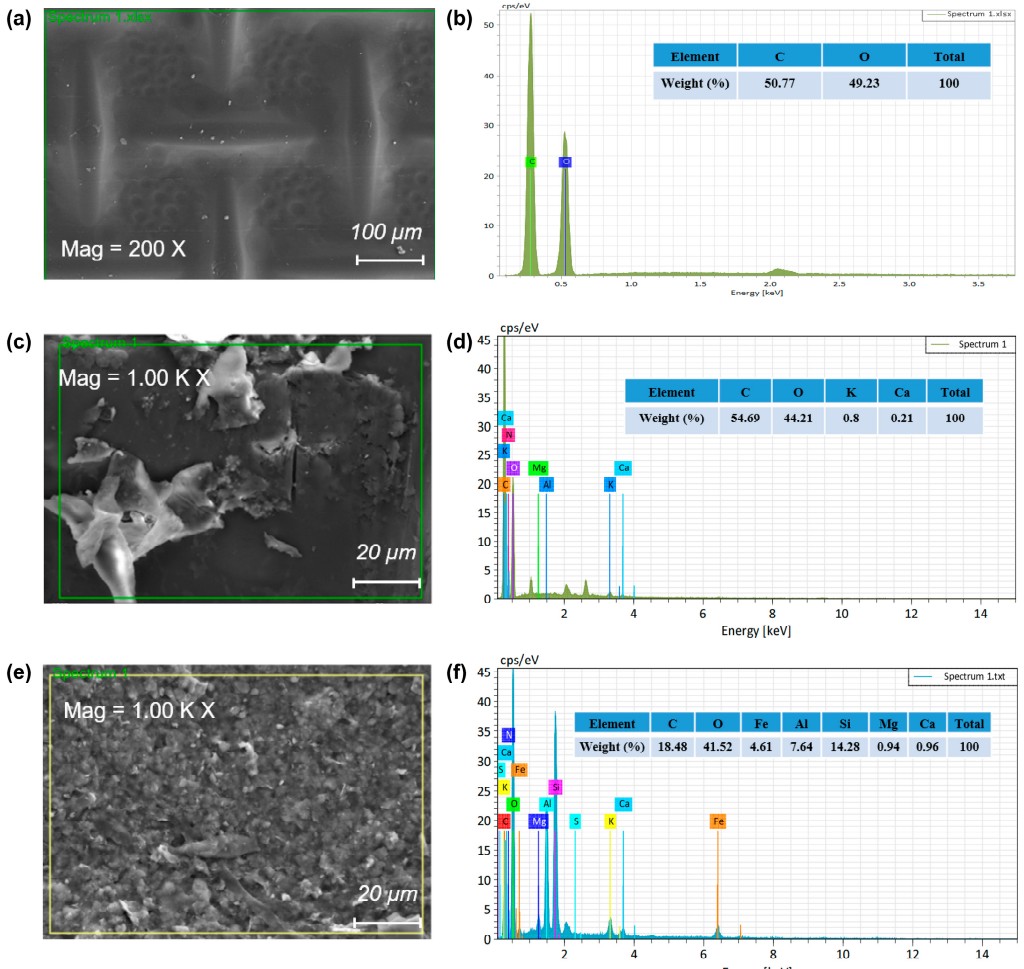

**Figure 7.** SEM–EDS analysis of the CTA membrane: unpolluted membrane (**a,b**); polluted active layer of membrane (**c,d**); and polluted support layer of membrane (**e,f**).

*3.4. Three-Dimensional Fluorescence Spectrum*

Figure 8 displays the fluorescence characteristics of dissolved organic matter in the DS and the FS, which helps to further explore the performance of FO in treating grass carp wastewater. By using the method of integration area, similar fluorescence peaks in the spectrum were divided into five representative zones: fluorescence zone I: Ex/Em = 200~250/200~300 nm; fluorescence zone II: Ex/Em = 200~250/300~400 nm; fluorescence zone III: Ex/Em = 200~250/400~600 nm; fluorescence zone IV: Ex/Em = 250~450/400~600 nm; and fluorescence zone V: Ex/Em = 250~450/200~400 nm. In accordance with Figure 8, the fluorescence center was mainly determined as zone I to zone V. Although zone V was the strongest fluorescent zone, zone I and zone IV were sub-strong fluorescent zones. The fluorescent substances in zones I, II, and V were related to polysaccharides in raw grass carp sewage, and the fluorescent substances in zones III and IV were mainly related to protein-like substances.

Figure 8a,c shows that the dissolved organic matter contained in the DS (sucrose solution) at the beginning of the running cycle of FO was significantly different from that at the end of the running cycle. In theory, the composition of the DS (sucrose extraction solution) of the FO system should remain the same at the beginning and the end of the experiment, but areas I and II increased significantly after the FO treatment, suggesting that the organic matter of the FS could penetrate into the DS. Comparing Figure 8a,c with Figure 8b,d, there is an obvious increase in zone V in Figure 8d, showing that the solute in the sucrose solution could also pass through the FO membrane and enter the FS, indicating that the sucrose solution might also lead to the problem of solute reverse osmosis.

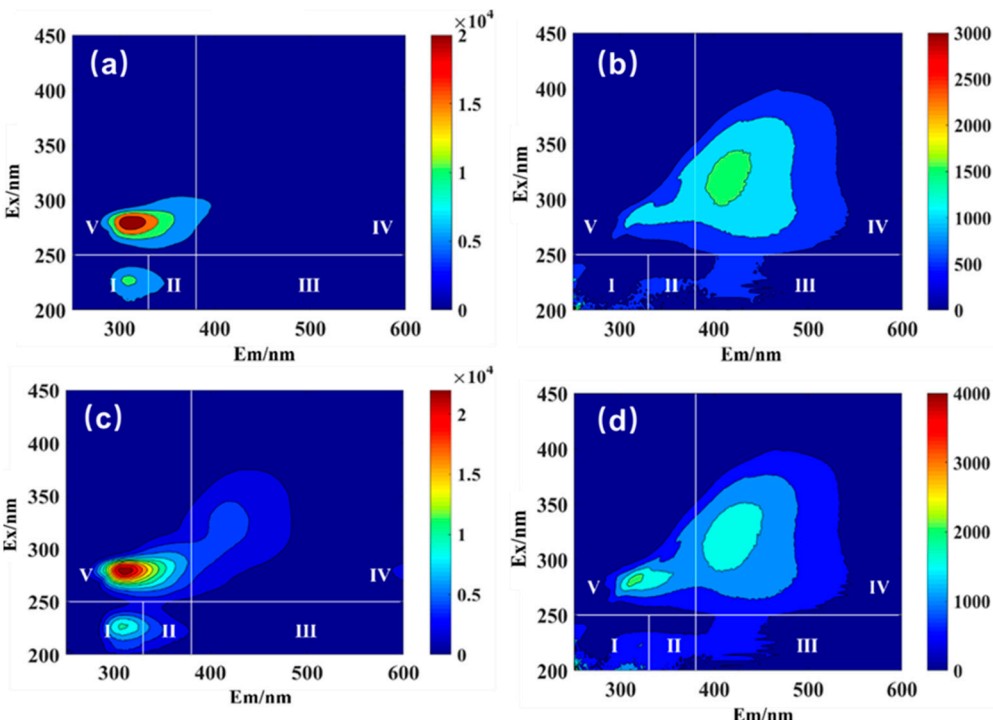

**Figure 8.** Three-dimensional fluorescence spectra of DS and FS: the DS at the beginning of the 20th cycle (**a**); the FS at the beginning of the 20th cycle (**b**); the DS at the end of the 20th cycle (**c**); and the FS at the end of the 20th cycle (**d**).

### 3.5. Cost and Energy Consumption

This section analyzes the operational cost of handling grass carp aquaculture wastewater by FO in this work, such as the consumption of FO membrane, sucrose solute, and electricity. The total cost of, and the total energy consumed in, the FO process are displayed in Table 2. In each experiment, the volume of grass carp pond wastewater was 1 L, the price of the CTA membrane was USD 606 m$^{-2}$, and the effective area of the membrane was 0.01 m$^2$. The reagent consumed in batch experiments was mainly the sucrose used to formulate the DS. In the real application, the economic cost of a ton of sucrose was USD 681 (USD 0.68 kg$^{-1}$). During the FO experiments, the concentration of the sucrose solution was 2 mol/L. In the whole 20 experimental batches, the reverse solute flux was 6.5 g·m$^{-2}$·h$^{-1}$ and the effective area of the membrane was 0.01 m$^2$. The sucrose reverse osmosis loss in every batch was 0.195 g, and the cost was USD 0.00013 based on the market price. At the same time, the device required two pumps, which consumed 15 w to work for 3 h. The electricity price was estimated at USD 0.1 kWh$^{-1}$. Therefore, the electricity cost for a membrane concentration setup of grass carp pond wastewater with this FO system was about USD 0.0045. The volume of the DS was 1 L, and calculation showed that the electric power consumed was 0.0123 kw·h, the electricity price was estimated at USD 0.1 kWh$^{-1}$, and the heating electricity fee was USD 0.0012.

**Table 2.** Operational costs in the pilot-scale experiment setup.

| Project | Cost/USD (1 T of Wastewater) |
| --- | --- |
| Sucrose loss | 0.13 |
| Pumps | 4.5 |
| Heating DS | 1.2 |
| Total | 5.83 |

In summary, considering the sucrose consumption, the operating power, the heating for the DS, etc., the total cost of treating 1-L grass carp wastewater was USD 0.00583 and

the total cost of treating 1-ton grass carp wastewater was estimated to be USD 5.83. Even though this could be considered a bit high, the operation cost could be reduced further, especially for the membrane part, when the technology is optimized in the future.

Table 3 compares the cost of the FO process used in this study with the costs of other kinds of FO wastewater treatment processes. An analysis of the energy consumed in different FO procedures shows that the total operating costs are determined by the physical and chemical characteristics of the wastewater to be treated, the concentration of the DS, the membrane cost, etc. Energy cost may increase with an increase in the complexity of the wastewater contaminants, an increase in the amount of solute in the DS, and an increase in the price of membrane materials. Local electricity price must also be considered when calculating cost. Hence, the economic control of FO treatment is a systematical task and it is worth making the effort to reduce energy consumption during the FO practice.

**Table 3.** Comparison of operational costs in wastewater treatment by the FO process.

| Feed Solution | Draw Solution | Water Flux (L/(m²·h)) | Treatment Costs | Reference |
|---|---|---|---|---|
| Aquaculture wastewater | 2 mol/L sucrose | 15.4–25.3 | USD 5.83 for 1 T | This work |
| Leachate | 1 mol/L NaCl | 1.5–6.7 | $0.276 \pm 0.033$ kWh/m$^3$ | [59] |
| DI water | Commercial liquid fertilizer | 5.7–14 | 0.09 kWh/m$^3$ | [60] |
| Wastewater from construction site | 0.6 mol/L NaCl | 3.9 | 7.88 USD/day | [61] |

## 4. Conclusions

(1) In this research, the treatment of grass carp pond wastewater via FO using sucrose solution as the DS was studied. The concentration of COD, the water flux, and the reverse solute flux were monitored in all the 20 batches of the experiment. The results indicated that the reverse solute flux and the water flux had a similar trend in batch experiments 1–8, suggesting that the reverse solute flux increased with the growth of water flux and vice versa.

(2) A loose cake layer formed on the membrane surface after the wastewater treatment process was completed. Physical cleaning improved the FO membrane system's water flux and the FO membrane selectivity efficiency. However, they could not recover to the previous initial level.

(3) The main elements in the EDS analysis of the FO membrane supportive layer were C, O, and Si, indicating that the membrane pollutants were mainly organic compounds and silicon dioxide. In addition to C and O, a small amount of K, Ca, and other trace elements were found via EDS detection.

(4) Three-dimensional fluorescence spectra of the FS and the DS led to the conclusion that the components of the FS could enter the DS through the membrane and the chemical components of the sucrose solution could also enter the FS, confirming the existence of reverse solute osmosis.

**Author Contributions:** Conceptualization, writing—original draft, validation and formal analysis, Y.X.; writing—review and editing, investigation and data curation, X.C.; methodology, resources, and funding acquisition, J.D. All authors have read and agreed to the published version of the manuscript.

**Funding:** This research received no external funding, and the APC was funded by Jianghui Du.

**Institutional Review Board Statement:** Not applicable.

**Informed Consent Statement:** Not applicable.

**Data Availability Statement:** Data are contained within the article.

**Conflicts of Interest:** The authors declare no conflict of interest.

## Abbreviations

| | |
|---|---|
| FO | Forward osmosis |
| FS | Feed solution |
| DS | Draw solution |
| COD | Chemical oxygen demand |
| SEM | Scanning electron microscope |
| EDS | Energy dispersive scanning |
| MD | Membrane distillation |
| NF | Nanofiltration |
| RO | Reverse osmosis |
| 3D | Three dimensional |
| RSM | Response surface methodology |
| AR | Analytical reagent |
| DI | Deionized |
| CTA | Cellulose triacetate |

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
