# Peer review of "Treatment of Wastewater from a Grass Carp Pond with Multiple-Batch Forward Osmosis by Using Sucrose as a Draw Solution"

_sustainability, doi:10.3390/su141610329_

Round 1

Reviewer 1 Report

This manuscript provides a sucrose drived FO process to treat grass carp ponds wastewater. The COD variation multiple, water flux, and reverse solute flux during the FO process were investigated. The process is well investigated and characterizations are adequate. I think this work is interesting and will appealing the reader‘s interest. So the manuscript is recommended to be accepted after minor revision.

1.     NH4+-N is easy to pass dense membrane. So what is the effect of the leakage of NH4+-N to the FO process? Could you provide more information?

2.     Compared to tradition FO process, authors provided the details about the cost and energy consumption. From the economy point of view, could you also provide the cost comparison in the manuscript about the using of traditional salts.

3.     Also, could you also provide a comparison about the flux and leakage of solute to highlight the merits of sucrose?

Author Response

We gratefully thank you for your time in making constructive remarks and useful suggestions!

Reviewer 2 Report

This study explored the treatment for wastewater from grass carp pond with multiple 2 batches forward osmosis by using sucrose as draw solution. As the topic is very specific, so needs to be justified among already available papers. The authors must address the below concerns before this manuscript can be considered for publication.

Some comments are:

The authors are talking about the RSM as the design of the experiment method but not mentioned the details. Explain.

Why authors used the commercially available membrane instead of the laboratory fabricated one?

What is the purpose of selecting 20 batch experiments? 

The authors did not mention the feed components and compositions. Add this information to the manuscript.

What is RMB?

Add a table of abbreviations.

Without mentioning the membrane properties how the membrane fouling study can be done.

Add membrane properties.

Mention the grass carp wastewater treatment with other membrane separation processes and compare it with the current work.

Author Response

(The authors gave the same response as above.)

Author Response

(The authors gave the same response as above.)

Round 2

Reviewer 2 Report

Accept in present form

Reviewer 3 Report

Thanks to all aouthers for providing the correct reply.

After the aouthors response I am agree to accept this manuscript for publication in sustainability journal.